# WHAT WOULD $\pi^*$ DO?: IMITATION LEARNING VIA OFF-POLICY REINFORCEMENT LEARNING

## ABSTRACT

Learning to imitate expert actions given demonstrations containing image observations is a difficult problem in robotic control. The key challenge is generalizing behavior to out-of-distribution states that differ from those in the demonstrations. State-of-the-art imitation learning algorithms perform well in environments with low-dimensional observations, but typically involve adversarial optimization procedures, which can be difficult to use with high-dimensional image observations. We propose a remarkably simple alternative based on off-policy soft Q-learning, which we call *soft Q imitation learning* (SQIL, pronounced "skill"), that rewards the agent for matching demonstrated actions in demonstrated states. The key idea is initially filling the agent's experience replay buffer with demonstrations, where rewards are set to a positive constant, and setting rewards to zero in all additional experiences. We derive SQIL from first principles as a method for performing approximate inference under the MaxCausalEnt model of expert behavior. The approximate inference objective trades off between a pure behavioral cloning loss and a regularization term that incorporates information about state transitions via the soft Bellman error. Our experiments show that SQIL matches the state of the art in low-dimensional environments, and significantly outperforms prior work in playing video games from high-dimensional images.

## 1 INTRODUCTION

Many real-world sequential decision-making problems can be tackled by imitation learning, where an expert demonstrates near-optimal behavior to an agent that then attempts to replicate that behavior in novel situations (Argall et al., 2009). This paper considers the problem of training an agent to imitate an expert policy, given expert action demonstrations and access to the environment. The agent does not get to observe a reward signal or query the expert, and does not know the state transition dynamics ex-ante.

Standard approaches to this problem based on behavioral cloning seek to imitate the expert's actions, but do not reason about the consequences of actions (Pomerleau, 1991). As a result, they suffer from state distribution shift, and fail to generalize to states that are very different from those seen in the demonstrations (Ross & Bagnell, 2010; Ross et al., 2011). Approaches based on inverse reinforcement learning (IRL) deal with this issue by fitting a reward function that represents preferences over trajectories rather than individual actions (Ng et al., 2000; Ziebart et al., 2008), and using the learned reward function to train the imitation agent through RL (Wulfmeier et al., 2015; Finn et al., 2016; Fu et al., 2017). This is the core idea behind generative adversarial imitation learning (GAIL), which implicitly combines IRL and RL using generative adversarial networks (GANs) (Ho & Ermon, 2016; Goodfellow et al., 2014). GAIL is state-of-the-art in environments with low-dimensional observations, but requires additional reward augmentation and feature engineering when applied to environments with high-dimensional image observations (Li et al., 2017), due to the difficulty of training GANs (Kurach et al., 2018). In short, imitation learning from raw pixel inputs without prior knowledge of the domain remains a challenge.

The key insight in this paper is that instead of using a learned reward function or discriminator to provide a reward signal to the imitation policy, as is done in IRL-based imitation methods and GAIL, the imitation agent can simply be rewarded for matching demonstrated actions in demonstrated states using off-policy soft Q-learning. The agent's experience replay buffer is initially filled

with demonstrations where rewards are set to some positive constant, and gradually accumulates additional experiences with rewards set to zero. The agent has an incentive to imitate the expert in demonstrated states, and to take actions that lead it back to demonstrated states when it encounters new states that differ from the demonstrations. We call this algorithm *soft Q imitation learning* (SQIL, pronounced "skill").

We motivate the SQIL algorithm by deriving it from first principles as a method for performing approximate inference under the MaxCausalEnt model of expert behavior (Ziebart et al., 2010). Our analysis shows that SQIL maximizes a lower bound on the log-posterior of the soft Q values given the demonstrations – a bound that trades off between a pure behavioral cloning loss and a regularization term that incorporates state transition information via the soft Bellman error.

The main contribution of this paper is SQIL: a practical and general imitation learning algorithm that is effective in MDPs with high-dimensional, continuous observations – including raw pixel inputs – and has a natural theoretical interpretation as approximate inference. We run experiments in the image-based Car Racing and low-dimensional Lunar Lander game environments from OpenAI Gym, the Humanoid and HalfCheetah continuous control tasks from MuJoCo, and image-based Atari Pong, to compare SQIL to two prior methods: behavioral cloning and GAIL. The results show that SQIL matches GAIL in low-dimensional environments, and significantly outperforms GAIL on the image-based Car Racing and Pong tasks.

## 2 PRELIMINARIES

This work builds on the maximum causal entropy (MaxCausalEnt) model of expert behavior (Ziebart et al., 2010; Levine, 2018). In a finite-horizon Markov Decision Process (MDP) with a finite, discrete action space $\mathcal{A}$,[1] the demonstrator is assumed to follow a policy $\pi$ that maximizes reward $R(s, a)$. To simplify exposition, we assume the state transition dynamics are deterministic: let $s'$ denote the state that follows from taking action $a$ in state $s$. The policy $\pi$ forms a Boltzmann distribution over actions,

$$\pi(a|s) \triangleq \exp\left(Q(s, a) - V(s)\right), \tag{1}$$

where $Q$ is the soft Q function, and $V$ is the soft value function,

$$V(s) \triangleq \begin{cases} 0 & \text{if } s \text{ is a terminal state} \\ \log\left(\sum_{a \in \mathcal{A}} \exp\left(Q(s, a)\right)\right) & \text{otherwise} \end{cases} \tag{2}$$

In the standard MaxCausalEnt model, the soft Q values are a deterministic function of the rewards and dynamics, given by the soft Bellman equation,

$$Q(s, a) \triangleq R(s, a) + \gamma V(s'). \tag{3}$$

### 2.1 APPROXIMATE MAXCAUSALENT MODEL

In the analysis in Section 3, the standard MaxCausalEnt model is approximated by a probabilistic model that treats the soft Q values as random variables,

$$Q(s, a) \sim \mathcal{N}(R(s, a) + \gamma V(s'), \sigma^2), \tag{4}$$

where $\sigma \in \mathbb{R}$ is a constant hyperparameter, and $R(s, a)$ is a random variable with zero mean and unit variance. To simplify the analysis in Section 3, we assume that the dynamics of the environment satisfy the property that if state $s_i$ is reachable from state $s_j$, then $s_j$ is not reachable from $s_i$ – this ensures that the probabilistic graphical model that captures the joint distribution over soft Q values is acyclic. The experiments in Section 5 show that even when this assumption is violated, SQIL performs well empirically.

---

[1]Assuming a discrete action space simplifies exposition. The analysis can be extended to continuous action spaces, as long as the entropy of policy $\pi(\cdot|s)$ is bounded above by some constant (see Equation 13), where $\pi$ is any policy in the policy class being optimized over. The SQIL algorithm can be applied to continuous control tasks using existing soft Q-learning methods (Haarnoja et al., 2017; 2018), as illustrated in Section 5.4.

# 3 IMITATION LEARNING VIA APPROXIMATE MAXCAUSALENT INFERENCE

We aim for an imitation learning algorithm that generalizes to new states, without resorting to adversarial optimization procedures that would be difficult to use with image observations. Our approach is to derive a lower bound on the log-posterior of the soft Q values given the demonstrations; in particular, a lower bound that is maximized by the SQIL algorithm.

## 3.1 DERIVING THE APPROXIMATE INFERENCE OBJECTIVE

Consider a demonstration rollout $\tau = (s_0, a_0, s_1, a_1, ..., s_T)$, where $s_T$ is a terminal state. Standard MaxCausalEnt IRL searches for the reward function that maximizes the posterior probability of the rewards given the demonstration. Our goal, however, is not to infer the expert's reward function; it is to infer the expert's policy, which is represented by their soft Q function. We approach this problem by formalizing uncertainty about the soft Q values using the probabilistic model described in Section 2.1. We fit a soft Q function to optimize the posterior probability of the soft Q values $\boldsymbol{Q}$ given the demonstration $\tau$, where $\boldsymbol{Q} \triangleq \{Q(s, a) \mid (s, a) \in \mathcal{X}\}$, and $\mathcal{X}$ is a finite subset of $\mathcal{S} \times \mathcal{A}$ that includes the state-action pairs in $\tau$. Since $Q(s, a)$ may depend on $Q(s', a')$ (see Equations 4 and 2), $\mathcal{X}$ is assumed to satisfy the closure property,

$$\forall(s, a) \in \mathcal{X}. \ (s' \text{ not terminal} \implies (\forall a' \in \mathcal{A}. \ (s', a') \in \mathcal{X})). \tag{5}$$

The goal is to derive a lower bound on the log-posterior, $\log p(\boldsymbol{Q}|\tau)$, that is maximized by the SQIL algorithm. The first step is to marginalize out the rewards, $\boldsymbol{R} \triangleq \{R(s, a) \mid (s, a) \in \mathcal{X}\}$, as follows.

$$p(\boldsymbol{Q}|\tau) \propto p(\tau|\boldsymbol{Q})p(\boldsymbol{Q}) = \mathbb{E}_{\boldsymbol{R}}[p(\tau|\boldsymbol{Q}, \boldsymbol{R})p(\boldsymbol{Q}|\boldsymbol{R})] = \mathbb{E}_{\boldsymbol{R}}[p(\tau|\boldsymbol{Q})p(\boldsymbol{Q}|\boldsymbol{R})]. \tag{6}$$

**First bound.** Optimizing the log of the expectation in Equation 6 is hard, for a number of reasons: we haven't assumed a particular prior distribution over rewards (only that it has zero mean and unit variance), and the soft Q values in $\boldsymbol{Q}$ are not conditionally independent given $\boldsymbol{R}$. Optimizing the expectation of the log is easier. By Jensen's inequality,

$$\log\left(\mathbb{E}_{\boldsymbol{R}}[p(\tau|\boldsymbol{Q})p(\boldsymbol{Q}|\boldsymbol{R})]\right) \geq \mathbb{E}_{\boldsymbol{R}}[\log p(\tau|\boldsymbol{Q}) + \log p(\boldsymbol{Q}|\boldsymbol{R})]. \tag{7}$$

The conditional likelihood of the demonstration $\tau$ given the soft Q values $\boldsymbol{Q}$ is given by Equation 1:

$$p(\tau|\boldsymbol{Q}) = \exp\left(\sum_{t=0}^{T-1} Q(s_t, a_t) - V(s_t)\right). \tag{8}$$

The conditional likelihood of the soft Q values $\boldsymbol{Q}$ given the rewards $\boldsymbol{R}$ is given by Equation 4:

$$p(\boldsymbol{Q}|\boldsymbol{R}) = \prod_{(s,a)\in\mathcal{X}} \mathcal{N}(Q(s, a); R(s, a) + \gamma V(s'), \sigma^2), \tag{9}$$

where the joint distribution over $\boldsymbol{Q}$ can be factored due to the state re-entry assumption about the dynamics made in Section 2.1 and the closure assumption about $\mathcal{X}$ made in Equation 5. Simplifying the lower bound in Equation 7,

$$(7) = \sum_{t=0}^{T-1} Q(s_t, a_t) - V(s_t)$$
$$+ \sum_{(s,a)\in\mathcal{X}} -\frac{1}{2\sigma^2}\left((Q(s, a) - (\mathbb{E}[R(s, a)] + \gamma V(s')))^2 + \text{Var}[R(s, a)]\right) - \log\sqrt{2\pi\sigma^2}. \tag{10}$$

Applying the assumption from Section 2.1 that the rewards have zero mean and unit variance,

$$(10) = \sum_{t=0}^{T-1} Q(s_t, a_t) - V(s_t) + \sum_{(s,a)\in\mathcal{X}} -\frac{1}{2\sigma^2}((Q(s, a) - \gamma V(s'))^2 + 1) - \log\sqrt{2\pi\sigma^2}. \tag{11}$$

**Second bound.** Next, the bound is intentionally loosened, in order to arrive at an objective that is maximized by the SQIL algorithm.[2] By the MaxCausalEnt assumptions in Section 2,

$$V(s_0) = \mathbb{E}_{\tau\sim p(\tau|\pi)}\left[\sum_{t=0}^{T-1} \gamma^t \left(Q(s_t, a_t) - \gamma V(s_{t+1}) + \mathcal{H}(\pi(\cdot|s_t))\right)\right], \tag{12}$$

---

[2]The experiments in Section 5 compare the performance of SQIL, which optimizes the bound in Equation 13, to the performance of SQIL-T, which optimizes the tighter bound in Equation 11. See Section A.1 in the appendix for a more detailed description of SQIL-T.

where $\mathcal{H}$ denotes entropy. Maximizing Equation 11 with respect to $Q$ yields a solution in which $Q(s,a) - \gamma V(s') \leq f(\sigma^2)$, where $f$ is an increasing function of $\sigma^2$. Thus, $V(s_0) \leq (f(\sigma^2) + \log|\mathcal{A}|)/(1-\gamma)$, and

$$(11) \geq (11) + V(s_0) - (f(\sigma^2) + \log|\mathcal{A}|)/(1-\gamma). \tag{13}$$

The soft Q function that maximizes the lower bound in Equation 13 is

$$Q^* \triangleq \arg\max_Q V(s_0) + \sum_{t=0}^{T-1} Q(s_t, a_t) - V(s_t) - \frac{1}{2\sigma^2} \sum_{(s,a)\in\mathcal{X}} (Q(s,a) - \gamma V(s'))^2. \tag{14}$$

**Practical implementation.** For environments with a small, discrete state space $\mathcal{S}$ and known dynamics, $\mathcal{X}$ can be set to $\mathcal{S} \times \mathcal{A}$. For environments with a continuous state space $\mathcal{S}$ and unknown dynamics, we set $\mathcal{X}$ to be the set of state-action pairs in the demonstration rollouts $\mathcal{D}_{\text{demo}}$ and rollouts $\mathcal{D}_{\text{samp}}$ of the imitation policy, which are periodically sampled during training.[3] We fit a parameterized soft Q function $Q_{\boldsymbol{\theta}}$ to minimize the loss,

$$\ell(\boldsymbol{\theta}) \triangleq \alpha \sum_{\tau\in\mathcal{D}_{\text{demo}}} \left( -V_{\boldsymbol{\theta}}(s_0) + \sum_{t=0}^{T-1} -(Q_{\boldsymbol{\theta}}(s_t, a_t) - V_{\boldsymbol{\theta}}(s_t)) \right) + \lambda_{\text{demo}} B_{\boldsymbol{\theta}}(\mathcal{D}_{\text{demo}}, 0) + \lambda_{\text{samp}} B_{\boldsymbol{\theta}}(\mathcal{D}_{\text{samp}}, 0), \tag{15}$$

where $\alpha, \lambda_{\text{demo}}, \lambda_{\text{samp}} \in \mathbb{R}_{\geq 0}$ are constant hyperparameters, $V_{\boldsymbol{\theta}}$ denotes the soft value function given by $Q_{\boldsymbol{\theta}}$ and Equation 2,[4] and $B_{\boldsymbol{\theta}}$ denotes the sum of squared soft Bellman errors,

$$B_{\boldsymbol{\theta}}(\mathcal{D}, r) \triangleq \sum_{\tau\in\mathcal{D}} \sum_{t=0}^{T-1} (Q_{\boldsymbol{\theta}}(s_t, a_t) - (r + \gamma V_{\boldsymbol{\theta}}(s_{t+1})))^2, \tag{16}$$

where $r \in \mathbb{R}$ is a constant that does not depend on the state or action. The experiments in Section 5 use a convolutional neural network or multi-layer perceptron to model $Q_{\boldsymbol{\theta}}$, where $\boldsymbol{\theta}$ are the weights of the neural network (Schmidhuber, 2015).

The loss in Equation 15 bounds the log-posterior of the soft Q values given the demonstrations. Searching for a soft Q function that optimizes this objective is equivalent to performing approximate inference under the MaxCausalEnt model of expert behavior. The objective combines a behavioral cloning loss, which does not reason about the consequences of actions, with a regularization term that incorporates information about state transitions. The regularization term performs soft Bellman backups on a reward of zero – a consequence of the prior distribution over rewards, which has an expected value of zero – and can be seen as imposing a prior distribution on the soft Q values that is independent of the expert demonstrations.

## 3.2 Implementing Approximate Inference with Off-Policy Soft Q-Learning

Surprisingly, the gradient of the approximate inference objective in Equation 15 is equivalent to the gradient of the objective of a soft Q-learning algorithm (Haarnoja et al., 2017) that gives the agent a constant positive reward for matching the demonstrated action in a demonstrated state, and zero reward otherwise. Differentiating the objective in Equation 15,

$$\nabla_{\boldsymbol{\theta}}\ell(\boldsymbol{\theta}) = \alpha \sum_{\tau\in\mathcal{D}_{\text{demo}}} \left( -\nabla V_{\boldsymbol{\theta}}(s_0) + \sum_{t=0}^{T-1} -(\nabla Q_{\boldsymbol{\theta}}(s_t, a_t) - \nabla V_{\boldsymbol{\theta}}(s_t)) \right)$$

$$+ \lambda_{\text{demo}} \sum_{\tau\in\mathcal{D}_{\text{demo}}} \sum_{t=0}^{T-1} \nabla(Q_{\boldsymbol{\theta}}(s_t, a_t) - \gamma V_{\boldsymbol{\theta}}(s_{t+1}))^2 + \lambda_{\text{samp}}\nabla B_{\boldsymbol{\theta}}(\mathcal{D}_{\text{samp}}, 0)$$

$$= \sum_{\tau\in\mathcal{D}_{\text{demo}}} \left( -\alpha\nabla V_{\boldsymbol{\theta}}(s_0) + \sum_{t=0}^{T-1} \alpha\nabla V_{\boldsymbol{\theta}}(s_t) - \alpha\gamma\nabla V_{\boldsymbol{\theta}}(s_{t+1}) \right)$$

$$+ \lambda_{\text{demo}}\nabla B_{\boldsymbol{\theta}}\left(\mathcal{D}_{\text{demo}}, \frac{\alpha}{2\lambda_{\text{demo}}}\right) + \lambda_{\text{samp}}\nabla B_{\boldsymbol{\theta}}(\mathcal{D}_{\text{samp}}, 0). \tag{17}$$

---

[3]The demonstrations alone may not satisfy the closure property in Equation 5. Augmenting the demonstrations with rollouts of the imitation policy improves our approximation of the objective in Equation 14.

[4]Note that there is a Q network $Q_{\boldsymbol{\theta}}$ and a value function $V$ that is defined in terms of $Q_{\boldsymbol{\theta}}$ (see Equation 2), rather than a separate Q network and V network.

---

**Algorithm 1** Soft Q Imitation Learning (SQIL, pronounced "skill")

---
1: Require $r, \lambda_{\text{samp}} \in \mathbb{R}_{\geq 0}, k \in \mathbb{N}$
2: Initialize $\mathcal{D}_{\text{samp}} \leftarrow \emptyset$
3: **for** $i = 1, 2, ...$ **do**
4: $\quad \boldsymbol{\theta} \leftarrow \boldsymbol{\theta} - \eta \nabla_{\boldsymbol{\theta}}(B_{\boldsymbol{\theta}}(\mathcal{D}_{\text{demo}}, r) + \lambda_{\text{samp}} B_{\boldsymbol{\theta}}(\mathcal{D}_{\text{samp}}, 0))$ $\qquad\qquad$ ▷ See Equation 16
5: $\quad$ **if** $i \bmod k \equiv 0$ **then**
6: $\qquad$ Sample rollout $\tau = (s_0, a_0, s_1, a_1, ..., s_T)$ with imitation policy $\pi_{\boldsymbol{\theta}}(a_t|s_t)$ $\qquad$ ▷ See Equation 1
7: $\qquad \mathcal{D}_{\text{samp}} \leftarrow \mathcal{D}_{\text{samp}} \cup \{\tau\}$
8: $\quad$ **end if**
9: **end for**

---

Setting $\gamma \triangleq 1$ turns the inner sum in the first term into a telescoping sum, yielding

$$(17) = \nabla \left( \lambda_{\text{demo}} B_{\boldsymbol{\theta}} \left( \mathcal{D}_{\text{demo}}, \frac{\alpha}{2\lambda_{\text{demo}}} \right) + \lambda_{\text{samp}} B_{\boldsymbol{\theta}}(\mathcal{D}_{\text{samp}}, 0) \right). \tag{18}$$

Setting $\lambda_{\text{demo}} \triangleq 1$ and $\alpha \triangleq 2\lambda_{\text{demo}} r$ yields the gradient of a soft Q-learning algorithm that gives the agent a reward of $r$ for taking the demonstrated action in a demonstrated state, and zero reward otherwise. The agent's experience replay buffer consists of $\mathcal{D}_{\text{demo}} \cup \mathcal{D}_{\text{samp}}$. We call this algorithm *soft Q imitation learning* (SQIL, pronounced "skill").

### 3.3 SOFT Q IMITATION LEARNING

SQIL is summarized in Algorithm 1. It performs soft Q-learning with two important modifications: (1) it initially fills the agent's experience replay buffer with demonstrations, where the rewards are set to some positive constant (e.g., $r \triangleq 1$), and (2) as the agent interacts with the world and accumulates new experiences, it adds them to the replay buffer, and sets the rewards for these additional experiences to zero. The agent has an incentive to imitate the expert in states that are similar to those encountered in the demonstrations. During training, the agent may encounter out-of-distribution states that are very different from those in the demonstrations. In such states, the agent has an incentive to take actions that lead it back to demonstration states, where rewards are positive.

Since the number of demonstrations is fixed and the number of additional experiences grows over time, we balance the number of demonstration experiences and additional experiences sampled for each gradient step in line 4 of Algorithm 1. As the imitation policy learns to behave more like the expert, the state-action distributions of $\mathcal{D}_{\text{samp}}$ and $\mathcal{D}_{\text{demo}}$ grow closer, which causes the reward for taking the expert action in an expert state to decay from $r$ to an expected value of $\frac{r}{1+\lambda_{\text{samp}}}$. This reward decay has the effect of making the imitation policy more stochastic over time, and could potentially be countered by decreasing $\lambda_{\text{samp}}$ to zero over time, or decreasing the temperature parameter of the stochastic policy, but neither of these fixes were necessary in our experiments.

One potential issue with SQIL is that by setting the reward for taking demonstrated actions in demonstrated states to a positive constant, and implicitly setting rewards at terminal states to zero (see Equation 2), SQIL may encourage the agent to deviate from the expert trajectories in order to avoid terminating the episode, especially in environments with long horizons. This is also an issue with GAIL, and is addressed in Anonymous (2019) by explicitly learning the rewards at absorbing states. One potential way to fix this issue in SQIL is to set the rewards for demonstrated transitions into terminal states to be $\frac{r}{1-\gamma}$ instead of $r$. This effectively augments the demonstration data with infinitely many rewarding transitions at absorbing states. This was not necessary in our experiments, since the horizons were relatively short (on the order of 1000 steps).

## 4 RELATED WORK

SQIL is inspired by the inverse soft Q-learning (ISQL) algorithm for internal dynamics estimation (Reddy et al., 2018), in that the objective in Equation 15 combines a behavioral cloning loss with soft Bellman error penalty terms, which is similar to the ISQL objective. The details of the two methods and their motivations are, however, completely different. ISQL is an internal dynamics estimation algorithm, while SQIL is meant for imitation learning. ISQL assumes that the demonstrations include observations of the expert's reward signal, while SQIL does not.

SQIL also resembles the Deep Q-learning from Demonstrations (DQfD) (Hester et al., 2017) and Normalized Actor-Critic (NAC) algorithms (Gao et al., 2018), in that all three algorithms fill the agent's experience replay buffer with demonstrations and include an imitation loss in the agent's objective. The key difference between SQIL and these prior methods is that DQfD and NAC are RL algorithms that assume access to a reward signal, while SQIL is an imitation learning algorithm that does not require an extrinsic reward signal from the environment. Instead, SQIL automatically constructs a reward signal from the demonstrations.[5]

## 5 SIMULATION EXPERIMENTS

The purpose of our experiments is three-fold. The first objective is to compare SQIL to prior work; in particular, on tasks with high-dimensional image observations, where prior methods are difficult to use. The second objective is to show that SQIL can be successfully deployed in MDPs with continuous action spaces. The third objective is to understand which components of SQIL contribute most to its performance. To accomplish the first objective, we use the image-based Car Racing and Atari Pong games, as well as the low-dimensional Lunar Lander environment from OpenAI Gym (Brockman et al., 2016), to benchmark SQIL against GAIL and behavioral cloning (BC). SQIL consistently outperforms BC, matches GAIL on low-dimensional Lunar Lander, and outperforms GAIL on the image-based Car Racing and Pong tasks. To accomplish the second objective, we use the Humanoid and HalfCheetah continuous control tasks from MuJoCo (Todorov et al., 2012) to show that SQIL performs comparably to GAIL on low-dimensional problems with continuous actions. To accomplish the third objective, we use the Lunar Lander game to conduct an ablation study that tests various hypotheses about which components of SQIL contribute to its performance.

### 5.1 COMPARISON TO PRIOR METHODS ON LOW-DIMENSIONAL LUNAR LANDER

The experiments in this section use the low-dimensional Lunar Lander game to verify that SQIL performs well on a relatively easy task. To emphasize the gap between imitation methods that take into account the consequences of actions, like SQIL and GAIL, and methods that do not, like BC, we exacerbate the state distribution shift that usually occurs due to compounding errors, by training the imitation agents in an environment with a different initial state distribution $\mathcal{S}_0^{\text{train}}$ than that of the expert demonstrations $\mathcal{S}_0^{\text{demo}}$. This intervention explicitly tests the generalization capabilities of imitation learning, by placing the agent in start states $\mathcal{S}_0^{\text{train}}$ that are very different from those seen in the demonstrations.

**Experimental setup.** The Lunar Lander game is a navigation task with deterministic dynamics in which the objective is to land on the ground, without crashing or flying out of bounds, using two lateral thrusters and a main engine. The action space consists of six discrete actions for steering and firing the main engine. Low-dimensional state observations $s \in \mathbb{R}^9$ encode position, velocity, orientation, and the location of the landing site. Demonstrations are collected from an expert trained using deep Q-learning (Mnih et al., 2015). The goal of this experiment is to study not only how well each method can mimic the expert demonstrations, but also how well they can acquire policies that generalize effectively to new states. After all, the main benefit of actually learning a policy, as opposed to simply playing back the expert's actions, is to acquire a strategy that generalizes to new situations. Therefore, we manipulate the initial state distribution of the environment in which the imitation agents are trained and evaluated. The expert demonstrations are collected starting from the initial state distribution $\mathcal{S}_0^{\text{demo}}$. The imitation agents are then trained in the same environment, but starting from a different initial state distribution $\mathcal{S}_0^{\text{train}}$. To create $\mathcal{S}_0^{\text{train}}$ in the Lunar Lander game, the agent is placed in a starting position that is rarely visited in the demonstrations. The agents are evaluated by training and testing with $\mathcal{S}_0^{\text{demo}}$, as well as training and testing with $\mathcal{S}_0^{\text{train}}$ – both results are reported in Table 1 and Figure 1. We measure the agents' success rate in landing at the target without crashing, flying out of bounds, or running out of time.

---

[5]DQfD and NAC are demonstration-accelerated RL algorithms, which require knowledge of the reward function, while SQIL is an imitation learning algorithm that does not. Hence, when a reward signal is observed, SQIL has no advantage over DQfD or NAC, just as pure imitation learning has no advantage over demonstration-accelerated RL.

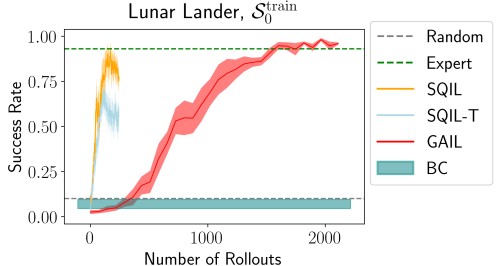

|          | $\mathcal{S}_0^{\mathrm{train}}$ | $\mathcal{S}_0^{\mathrm{demo}}$ |
|----------|--------------|--------------|
| Random   | $0.10 \pm 0.30$ | $0.04 \pm 0.02$ |
| BC       | $0.07 \pm 0.03$ | $\mathbf{0.93 \pm 0.03}$ |
| GAIL     | $\mathbf{0.98 \pm 0.01}$ | $\mathbf{0.95 \pm 0.02}$ |
| SQIL-T   | $0.66 \pm 0.02$ | $\mathbf{0.89 \pm 0.01}$ |
| SQIL     | $\mathbf{0.89 \pm 0.02}$ | $\mathbf{0.88 \pm 0.03}$ |
| Expert   | $0.93 \pm 0.03$ | $0.89 \pm 0.31$ |

Figure 1: Error regions show standard error across five random seeds. BC performance after training. X-axis represents amount of interaction with the environment (not expert demonstrations).

Table 1: Lunar Lander. Best performance on 100 consecutive episodes during training. Standard errors shown for five random seeds.

**Analysis.** The results show that when there is no variation in the initial state, all methods perform equally well (see the $\mathcal{S}_0^{\mathrm{demo}}$ column in Table 1). The task is easy enough that even behavioral cloning achieves a high success rate. When the initial state is changed for the purpose of testing generalization, SQIL and GAIL both perform much better than BC (see Figure 1 and the $\mathcal{S}_0^{\mathrm{train}}$ column in Table 1). This is unsurprising, since BC suffers from state distribution shift, and GAIL tends to perform well on tasks with low-dimensional observations.

GAIL achieves a higher success rate than SQIL, but requires many more interactions with the environment. This can most likely be attributed to the fact that GAIL uses TRPO (Schulman et al., 2015) for policy optimization, while SQIL uses Q-learning. Since this difference in the underlying RL algorithm makes it difficult to fairly compare SQIL to GAIL, the results are merely intended to illustrate that both SQIL and GAIL can outperform BC and perform on par with the expert. In the following two sections, we show that SQIL significantly outperforms GAIL on high-dimensional, image-based tasks.

SQIL, which optimizes the bound in Equation 13, performs better than its cousin, SQIL-T, which optimizes the tighter bound in Equation 11. This is a somewhat surprising result, since optimizing a tighter bound on the log-posterior should lead to better performance. One possible explanation for this not being the case is that in practice, optimizing Equation 11 via gradient descent is harder than optimizing Equation 13.

### 5.2 COMPARISON TO PRIOR METHODS ON IMAGE-BASED CAR RACING

Section 5.1 showed that SQIL performs on par with GAIL and better than BC in environments with low-dimensional observations. This section shows that SQIL can significantly outperform GAIL and BC in environments with high-dimensional image observations. With images, the policy learning problem becomes substantially harder: now, the policy must learn to interpret raw image observations and determine the best course of action. This is notoriously difficult for methods based on adversarial learning, such as GAIL.

**Experimental setup.** The Car Racing game is a navigation task with deterministic dynamics in which the objective is to visit as many road markers as possible, while avoiding the grass. The agent receives 64x64 RGB images as observations, and outputs a discrete steering command (left, right, or straight). An episode lasts at most 1000 steps, and terminates early if the car deviates too far from the road. Demonstrations are collected from an expert trained using model-based RL (Ha & Schmidhuber, 2018). The initial state distribution is manipulated as in Section 5.1. To create $\mathcal{S}_0^{\mathrm{train}}$ in the Car Racing game, the car is rotated 90 degrees so that it begins perpendicular to the track, instead of parallel to the track as in $\mathcal{S}_0^{\mathrm{demo}}$. This simple intervention presents a significant generalization challenge to the imitation learner, since the expert demonstrations do not contain any examples of states where the car is perpendicular to the road, or even significantly off the road axis. The agent must learn to make a tight turn to get back on the road, then stabilize its orientation so that it is parallel to the road, and only then proceed forward to mimic the expert demonstrations. We measure the reward collected by the agent, which counts the number of road markers it visits,

and includes a -100 penalty for deviating too far from the road and causing the episode to terminate early.

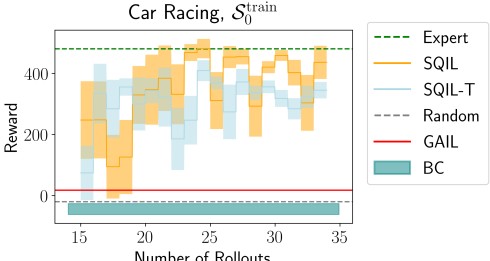

| | $\mathcal{S}_0^{\text{train}}$ | $\mathcal{S}_0^{\text{demo}}$ |
|---|---|---|
| Random | $-21 \pm 56$ | $-68 \pm 4$ |
| BC | $-45 \pm 18$ | $\mathbf{698 \pm 10}$ |
| GAIL | $17 \pm 0$ | $7 \pm 1$ |
| SQIL-T | $311 \pm 10$ | $\mathbf{693 \pm 7}$ |
| SQIL | $\mathbf{375 \pm 19}$ | $704 \pm 6$ |
| Expert | $480 \pm 11$ | $704 \pm 79$ |

Figure 2: Error regions show standard error across five random seeds. GAIL and BC performance after training.

Table 2: Car Racing. Final performance on 20 episodes after training. Standard errors shown for five random seeds.

**Analysis.** Table 2 and Figure 2 show that SQIL performs much better than both BC and GAIL, when starting from $\mathcal{S}_0^{\text{train}}$. SQIL learns to make a tight turn that takes the car through the grass and back onto the road, then stabilizes the car's orientation so that it is parallel to the track, and then proceeds forward like the expert does in the demonstrations. GAIL does not improve upon its initialized policy, which keeps turning in a loop, and does not move forward once the agent is parallel to the road. BC tends to drive straight ahead into the grass instead of turning back onto the road. One explanation of these results is that GAIL uses TRPO (Schulman et al., 2015) to fit the imitation policy, which inherits the challenges of RL with image observations. SQIL, on the other hand, inherits the ease of fitting an image-based policy from behavioral cloning via the analysis in Section 3, and only relies on soft Bellman backups for regularization and fine-tuning (vs. RL from scratch). SQIL outperforms SQIL-T, as in Section 5.1, though the performance gap is smaller here than in Section 5.1.

## 5.3 Comparison to Prior Methods on Image-Based Atari Pong

The experiments in the previous sections compare SQIL, which uses Q-learning, to GAIL, which uses TRPO. This can make it difficult to compare the performance and sample efficiency of SQIL and GAIL. The experiment in this section compares SQIL to a version of GAIL that uses Q-learning instead of TRPO, on the Pong Atari game with raw image observations. This allows for a head-to-head comparison of SQIL and GAIL: both algorithms use the same underlying RL algorithm (Q-learning) but provide the agent with different rewards – SQIL provides constant rewards, while GAIL provides learned rewards. The results in the figure to the right show that SQIL outperforms GAIL on this challenging image-based task.

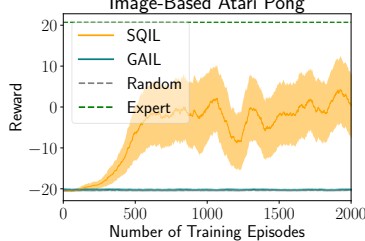

Smoothed with a rolling window of 100 episodes. Standard error on three random seeds.

## 5.4 Comparison to Prior Methods on Continuous Control Tasks

The experiments in the previous sections evaluate SQIL in environments with discrete action spaces. The experiments in this section illustrate how SQIL can be adapted to continuous action spaces. We instantiate SQIL using soft actor-critic (SAC) – an off-policy RL algorithm that can solve continuous con-

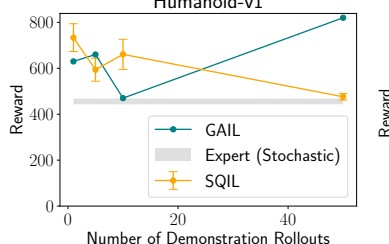
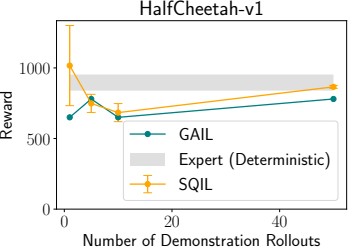

SQIL: best performance on 10 consecutive training episodes. GAIL: results from Dhariwal et al. (2017).

trol tasks (Haarnoja et al., 2018). In particular, SAC is modified in the following ways: (1) the agent's experience replay buffer is initially filled with expert demonstrations, where rewards are set to a positive constant, (2) when taking gradient steps to fit the agent's soft Q function, we sample a balanced number of demonstration experiences and new experiences from the replay buffer, and (3) the agent observes rewards of zero during their interactions with the environment, instead of an extrinsic reward signal that specifies the desired task. This instantiation of SQIL is compared to GAIL on the Humanoid (17 DoF) and HalfCheetah (6 DoF) tasks from MuJoCo (Todorov et al., 2012). The results in the figure to the right show that SQIL and GAIL perform comparably to each other on both Humanoid and HalfCheetah, demonstrating that SQIL can be successfully deployed on problems with continuous action spaces.

## 5.5 Ablation Study

The experiments in Sections 5.1 and 5.2 show that SQIL can outperform prior methods, but they do not illuminate how the various terms in the SQIL gradient in line 4 of Algorithm 1 interact to yield these results. The experiments in this section use the Lunar Lander environment to conduct an ablation study to better understand the importance of each component of SQIL. We hypothesize that incorporating the right information about state transitions into the agent's experience replay buffer is essential: the replay buffer needs to contain more than just the demonstrations, and these additional experiences need to be sampled on-policy (see line 6 of Algorithm 1).

**Experimental setup.** We manipulate the parameters of SQIL to hobble it in various ways that test our hypotheses. In the first condition, $\lambda_{\text{samp}}$ is set to zero, to prevent SQIL from using additional samples drawn from the training environment. In the second condition, $\gamma$ is set to zero to prevent SQIL from accessing information about state transitions. In the third condition, a uniform random policy is used to sample additional rollouts instead of the imitation policy $\pi_\theta$ in line 6 of Algorithm 1. We measure the agents' success rate, as in Section 5.1.

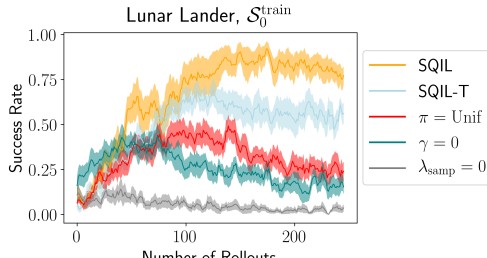

|  | $\mathcal{S}_0^{\text{train}}$ | $\mathcal{S}_0^{\text{demo}}$ |
|---|---|---|
| Random | $0.10 \pm 0.30$ | $0.04 \pm 0.02$ |
| $\lambda_{\text{samp}} = 0$ | $0.12 \pm 0.02$ | $\mathbf{0.87 \pm 0.02}$ |
| $\gamma = 0$ | $0.41 \pm 0.02$ | $\mathbf{0.84 \pm 0.02}$ |
| $\pi = \text{Unif}$ | $0.47 \pm 0.02$ | $\mathbf{0.82 \pm 0.02}$ |
| SQIL-T | $0.66 \pm 0.02$ | $\mathbf{0.89 \pm 0.01}$ |
| SQIL | $\mathbf{0.89 \pm 0.02}$ | $\mathbf{0.88 \pm 0.03}$ |
| Expert | $0.93 \pm 0.03$ | $0.89 \pm 0.31$ |

Figure 3: Error regions show standard error across five random seeds.

Table 3: Lunar Lander. Best performance on 100 consecutive episodes during training. Standard errors shown for five random seeds.

**Analysis.** The results in Figure 3 and Table 3 show that the original method performs significantly better than its hobbled counterparts when starting from $\mathcal{S}_0^{\text{train}}$, suggesting that SQIL relies on information about the environment dynamics encoded in state transitions (SQIL vs. $\gamma = 0$), and that SQIL requires on-policy sampling (SQIL vs. $\pi = \text{Unif}$ and $\lambda_{\text{samp}} = 0$).

## 6 Discussion

**Summary.** We contribute a practical and general algorithm for learning to imitate an expert given action demonstrations and access to the environment. Simulation experiments demonstrate the effectiveness of our method at recovering an imitation policy that performs well in comparison to GAIL and behavioral cloning on tasks with high-dimensional, continuous observations as well as continuous actions.

**Limitations.** The analysis in Section 3, which proves that SQIL is equivalent to approximate inference of the expert policy under the MaxCausalEnt model of expert behavior, is limited in that does not show that SQIL exactly recovers the expert policy in the limit of having infinite demonstrations, or that the optimal imitation policy's state-action occupancy measure converges to the

expert's. These are desirable properties for an imitation learning algorithm, and it is unclear if SQIL possesses them.

**Future work.** The ability to robustly recover an imitation policy from image observations, even when the initial state distribution differs between demonstration time and training time, enables open-world robotics experiments in which using sensors other than a camera would be expensive, and consistently resetting the environment to the same set of start states is difficult.

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

# A APPENDIX

## A.1 SQIL-T

SQIL-T uses Adam (Kingma & Ba, 2014) to minimize the loss function,

$$\ell_{\text{SQIL-T}}(\boldsymbol{\theta}) \triangleq \alpha \sum_{\tau \in \mathcal{D}_{\text{demo}}} \sum_{t=0}^{T-1} -(Q_{\boldsymbol{\theta}}(s_t, a_t) - V_{\boldsymbol{\theta}}(s_t)) + \lambda_{\text{demo}} B_{\boldsymbol{\theta}}(\mathcal{D}_{\text{demo}}, 0) + \lambda_{\text{samp}} B_{\boldsymbol{\theta}}(\mathcal{D}_{\text{samp}}, 0),$$

(19)

which is identical to the SQIL loss in Equation 15, except that the $V_{\boldsymbol{\theta}}(s_0)$ terms are not present in Equation 19.

## A.2 IMPLEMENTATION DETAILS

Code and data for the experiments in Section 5 are publicly available.[6]

To ensure fair comparisons, the same network architectures were used to evaluate SQIL, GAIL, and BC. For Lunar Lander, we used a network architecture with two fully-connected layers containing 128 hidden units each to represent the Q network in SQIL, the policy and discriminator networks in GAIL, and the policy network in BC. For Car Racing, we used four convolutional layers (following Ha & Schmidhuber (2018)) and two fully-connected layers containing 256 hidden units each. For Humanoid and HalfCheetah, we used two fully-connected layers containing 256 hidden units each. For Pong, we used the convolutional neural network described in Mnih et al. (2015) to represent the Q network in SQIL, as well as the Q network and discriminator network in GAIL.

To ensure fair comparisons, the same demonstration data was used to train SQIL, GAIL, and BC. For Lunar Lander, we collected 100 demonstration rollouts. For Car Racing, we collected 20 demonstration rollouts. For Pong, we collected 50 demonstration rollouts. Expert demonstrations were generated from scratch for Lunar Lander using DQN (Mnih et al., 2015), and collected from open-source pre-trained policies for Car Racing (Ha & Schmidhuber, 2018) as well as Humanoid and HalfCheetah (Dhariwal et al., 2017). The Humanoid demonstrations were generated by a stochastic expert policy, while the HalfCheetah demonstrations were generated by a deterministic expert policy; both experts were trained using TRPO.[7] We used two open-source implementations of GAIL: Fu et al. (2017) for Lunar Lander and Car Racing, and Dhariwal et al. (2017) for MuJoCo. We adapted the OpenAI Baselines implementation of GAIL to use Q-learning for Pong. Expert demonstrations were generated from scratch for Pong using DQN.

For Lunar Lander, we set $r = 1$, $k = 200$, and $\lambda_{\text{samp}} = 10^{-6}$. For Car Racing, we set $r = 1$, $k = 50$, and $\lambda_{\text{samp}} = 0.01$. For Humanoid, we set $r = 5$ and $\lambda_{\text{samp}} = 1$. For HalfCheetah, we set $r = 10$ and $\lambda_{\text{samp}} = 1$. For Pong, we set $r = 100$, $k = 1$, and $\lambda_{\text{samp}} = 1$.

SQIL was not pre-trained in any of the experiments. GAIL was pre-trained using behavioral cloning for HalfCheetah, but was not pre-trained in any other experiments.

In standard implementations of Q-learning and SAC, the agent's experience replay buffer typically has a fixed size, and once the buffer is full, old experiences are deleted to make room for new experiences. In SQIL, we never delete demonstration experiences from the replay buffer, but otherwise follow the standard implementation.

We use Adam (Kingma & Ba, 2014) to take the gradient step in line 4 of Algorithm 1.

The GAIL performance metrics in Section 5.4 are taken from Dhariwal et al. (2017).[8]

The GAIL and SQIL policies in Section 5.4 are set to be deterministic during the evaluation rollouts used to measure performance.

---

[6]https://drive.google.com/file/d/1fQpMza1-9wBmKYxfZ0bZvjw8rFBoVdgv/view?usp=sharing

[7]https://drive.google.com/drive/folders/1h3H4AY_ZBx08hz-Ct0Nxxus-V1melu1U

[8]https://github.com/openai/baselines/blob/master/baselines/gail/result/gail-result.md

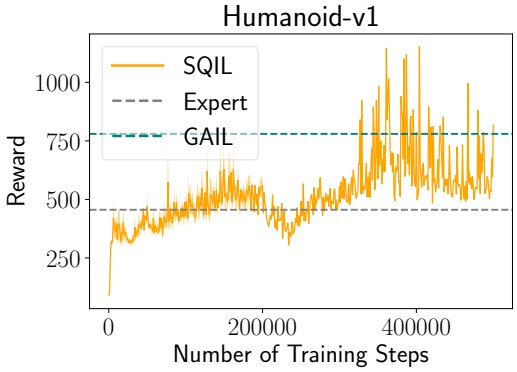

Figure 4: Standard error shown for five random seeds on the first 200k steps. No smoothing across training steps. The first four random seeds are from the experiment in Section 5.4, which only train for 200k steps. To measure the effect of training on more episodes, we ran another experiment with a fifth random seed and trained for 500k steps. Training for a much larger number of steps does not lead to significant deterioration in performance.

## A.3 RELATED WORK

Various approaches have been developed to address state distribution shift in behavioral cloning, without relying on IRL. Hand-engineering a domain-specific loss function and carefully designing the demonstration collection process have enabled researchers to train effective imitation policies for self-driving cars (Bojarski et al., 2016), autonomous drones (Giusti et al., 2016), and robotic manipulators (Zhang et al., 2017; Rahmatizadeh et al., 2017; 2016). DAgger-based methods query the expert for on-policy action labels (Ross et al., 2011; Laskey et al., 2016). These approaches either require domain knowledge or the ability to query the expert, while SQIL requires neither.

