# OpenReview forum: "What Would pi* Do?: Imitation Learning via Off-Policy Reinforcement Learning"
_ICLR.cc/2019/Conference_

### Official Review · AnonReviewer2 · 2018-11-01
**Interesting and promising idea that could be practical**

**Rating:** 6
**Confidence:** 4

**Review:**

The paper introduces a relatively simple method for imitation learning that seems to be successful despite its simplicity. The method, SQIL, assigns a constant positive reward (r) to the demonstrations and zero reward to generated trajectories. While I like the connections between SQIL and SQL and the simplicity of the idea, I think there are several issues which connections with GAIL that are not discussed; some "standard" environments (such as Mujoco) that SQIL has not compared against the baselines. I believe the paper would have a bigger impact after some addressing some of the issues.

(
Update: I am glad that the authors added updates to the experiments. I think the method could be practical due to the simplicity, therefore of interest to ICLR.

The Pong case is also quite interesting, although it seems slightly "unfair" since the true reward of Pong is also sparse and DQN could do well on it. I think the problem with GAIL is that the reward could be hard to learn in high-dimensional cases, so it is hard to find good hyperparameters for GAIL on the Pong case. This shows some potential of the idea behind using simple rewards.
)

1. The first issue is the similarity with GAIL in the "realistic" setting. Since we cannot have infinite expert demonstrations, there would always be some large enough network that could perfectly distinguish the demonstrations (assign reward to 1) and the generated policies (assign reward to 0). Therefore, it would seem to me that from this perspective SQIL is an instance of GAIL where the discriminator is powerful and expert demos are finite (and disjoint from generated trajectories, which is almost always the case for continuous control). In the finite capacity case, I am unsure whether the V and Q networks in SQIL does a similar job as the discriminator in GAIL / AIRL type algorithms, since both seem to extrapolate between demonstrations and generations?

2. Moreover, I don't think SQIL would always recover the expert policy even with infinite demonstrations. For example, lets think about the Reacher environment, where the agent controls a robotic arm to reach a target location. The expert demonstration is the fastest way of reaching the target (move minimum distance between joints). If we consider the MDP to have possibly very large / infinite horizon (only stops when it reaches the target), I could construct a hack policy that produces larger episodic reward compared to the expert. The policy would simply move back and forth between two expert demonstrated states, where it would receive 1 reward in the states for odd time and 0 reward for the states for even time. The reward would be something like 1 / (1 - \gamma^2) compared to the experts' reward which is \sum_{i=0..T} \gamma^{i} = (1 - \gamma^{T+1}) / (1 - \gamma).

Some fix would be to set the reward for generated policies to be negative, or introduce some absorbing state where the expert will still receive the positive reward even after reaching the target (but that is not included in demonstrations). Nevertheless, a suitable reward prior seems to be crucial to the success of this SQIL, as with GAIL requiring reward augmentation.

3. Despite the above issues, I think this could be a very practical method due to its (perhaps surprising) simplicity compared to GAIL. However, the experiments only considered two environments that are not typically considered by GAIL; I believe SQIL would make a bigger impact if it is compared with GAIL in Mujoco environments -- seems not very implementation heavy because your code is based on OpenAI baselines anyway. Mujoco with image inputs would also be relevant (see ACKTR paper).

Minor points:
- What is the underlying RL algorithm for GAIL? It would seem weird if you use Q-learning for SQIL and TRPO for GAIL, which makes it impossible to identify whether Q-learning or SQIL contributed more to the performance. While GAIL used TRPO in the original paper, it would be relatively straightforward to come up with some version of GAIL that uses Q-learning.
- Some more details in background for MaxEnt RL to make the paper more self contained.
- More details about the hyperparameters of SQIL in experiments -- e.g. what is \lambda?
- Did you pretrain the SQIL / GAIL policies? Either case, it would be important to mention that and be fair in terms of the comparison.
- Why does SQIL-11 perform worse than SQIL even though it is a tighter bound?
- wrt. math -- I think the anonymous comment addressed some of my concerns, but I have not read the updated version so cannot be sure.

---

> ### Author Response · Authors · 2018-11-13
> **Response to Reviewer 2**
>
> Thank you for the thoughtful review. We have addressed each of your suggestions, and added the following experiments: comparisons between SQIL and GAIL on the MuJoCo Humanoid and HalfCheetah tasks in Section 5.4, and on image-based Atari Pong in Section 5.3. We believe that this addresses all concerns, but we are happy to respond to any other suggestions you might have. We look forward to your response.
>
> > 3. Despite the above issues, I think this could be a very practical method due to its (perhaps surprising) simplicity compared to GAIL. However, the experiments only considered two environments that are not typically considered by GAIL...
>
> We have added MuJoCo experiments to Section 5.4, in which both SQIL and GAIL generally achieve good performance using a very small number of demonstrations. In this setting, SQIL does not hold an advantage over GAIL, since GAIL already performs near-optimally.
>
> > What is the underlying RL algorithm for GAIL? It would seem weird if you use Q-learning for SQIL and TRPO for GAIL...
>
> We have added a preliminary experiment on image-based Atari Pong to Section 5.3, in which we implement a version of GAIL that uses Q-learning instead of TRPO. This experiment provides a head-to-head comparison of SQIL and GAIL: both algorithms use the same underlying RL algorithm (Q-learning) but provide the agent with different rewards -- SQIL provides a constant reward, while GAIL provides a learned reward.
>
> In the other experiments, we use open-source implementations of GAIL that use TRPO, as in Ho et al. (2016). Two concurrent submissions to ICLR 2019 (https://openreview.net/pdf?id=BkN5UoAqF7 and https://openreview.net/forum?id=Hk4fpoA5Km) develop variants of GAIL that use off-policy RL (e.g., Q-learning) instead of TRPO to train the policy network, and contain results that show the gains from using off-policy RL over TRPO are mainly in sample efficiency with respect to the number of interactions with the environment, rather than the performance of the final imitation policy. In our experiments, we focus on measuring the performance of the final imitation policy.
>
> > 1. The first issue is the similarity with GAIL in the "realistic" setting...
>
> The reviewer raises a good point: SQIL is similar to a variant of GAIL in which the discriminator completely overfits to the demonstrations. However, there is no separate V network, only a Q network, and a V that is defined in terms of Q. We have added a footnote to the bottom of page 4 to clarify this. The Q network in SQIL is doing the job of both the policy and discriminator networks in GAIL.
>
> > 2. Moreover, I don't think SQIL would always recover the expert policy even with infinite demonstrations...
>
> The reviewer raises a good point. By setting the reward for taking demonstrated actions in demonstrated states to a *positive* constant, and implicitly setting rewards at absorbing states to zero, SQIL may encourage the agent to deviate from the expert trajectories in order to avoid terminating the episode in environments with long horizons. This is also an issue with GAIL, and is addressed by a concurrent submission to ICLR 2019: https://openreview.net/forum?id=Hk4fpoA5Km.
>
> One potential way to fix this issue in SQIL is to set the rewards for demonstrated transitions into absorbing states to be r/(1-gamma) instead of r. This effectively augments the demonstration data with infinitely many rewarding transitions at absorbing states. This was not necessary in our experiments, since the horizons were relatively short (on the order of ~1000 steps). We have clarified this in Section 3.3.
>
> > Some more details in background for MaxEnt RL to make the paper more self contained.
>
> We have revised Section 2 to clarify the approximate MaxCausalEnt model.
>
> > More details about the hyperparameters of SQIL in experiments -- e.g. what is \lambda?
>
> We have added implementation details to Section A.2 in the appendix, and anonymously released code (see Section A.1 in the appendix) to encourage transparency and reproducibility.
>
> > Did you pretrain the SQIL / GAIL policies? Either case, it would be important to mention that and be fair in terms of the comparison.
>
> The SQIL and GAIL policies were not pre-trained. We have included these and other implementation details in Section A.2 of the appendix.
>
> > wrt. math -- I think the anonymous comment addressed some of my concerns, but I have not read the updated version so cannot be sure.
>
> We have revised the analysis in Sections 2 and 3 to improve clarity.

---

> ### Author Response · Authors · 2018-11-17
> **Request for Suggestions for Improvement**
>
> Thank you for taking the time to read our response and update your review. Since Pong and other Atari games may not be the fairest tasks on which to compare SQIL and GAIL, could you suggest other domains where we could run further experiments? We would also appreciate any additional feedback on the revised paper. Are there any other aspects of the paper that you think could be improved?

---

> > ### Comment · AnonReviewer2 · 2018-11-25
> > **I think Pong is relatively fair actually in this setting.**
> >
> > I think Pong is relatively fair actually in this setting, I guess it does not require as much tuning as GAIL due to the fact that DQN is able to do well even with (relatively) sparse rewards.

---

> > > ### Author Response · Authors · 2018-11-25
> > > **Thank you for the clarification**
> > >
> > > Are there any other experiments or revisions you would like to see, that would affect your overall rating of the paper?

---

### Official Review · AnonReviewer1 · 2018-11-03

**Rating:** 5
**Confidence:** 3

**Review:**

This paper proposes an imitation learning algorithm framed as an off-policy RL scenario. They introduce all demonstrations into a replay buffer, with positive reward r=1. Subsequent states derived from the agent will be also introduced in the replay buffer but with reward r=0. There is no additional reward involved. The hope is that agents will learn to match the expert in states appearing in the replay buffer and not try to do much else.

Overall, I am not convinced that the paper accurately supports its claims.

1.	The authors support their method by saying that extending GAIL and other imitation learning algorithms to support pixel observations has failed. However, there are actually some papers showing quite successful applications of doing that: e.g. see Li et al. 2017 in challenging driving domain in TORCS simulator.

2.	More problematic, I think there is a clear flaw with this algorithm: imagine having trained successfully and experiencing many trajectories that are accurately reproducing the behaviour of the expert. Given their method, all these new trajectories will be introduced into the replay buffer with a reward of 0, given they come from the agent. What will the gradients be when provided with state-action pairs with both r=0 and r=1? These situations will have high variance (even though they should be clear r=1 situations) and this will hinder learning, which will tend to  decay the behaviour as time goes on.
This actually seems to be happening, see Figure 1 at the end: both SQIL curves appear to slowly starting to decay.
This is why GAIL is training its discriminator further, you want to keep updating the distribution of “agent” vs “expert”, I’m not sure how this step can be bypassed?

3.	How would you make sure that the agent even starts encountering rewarding states? Do you need deterministic environments where this is more likely to happen? Do you need conditions on how much of the space is spanned by the expert demonstrations?

4.	Additionally, Figure 1 indicates BC and GAIL as regions, without learning curves, is that after training?

5.	I am not convinced of the usefulness of the lengthy derivation, although I could not follow it deeply. Especially given that the lower bound they arrive at does not seem to accurately reflect the mismatch in distributions as explained above.

6.	There are no details about the network architecture used, about the size of the replay buffer, about how to insert/delete experience into the replay buffer, how the baselines were set up, etc. There are so few details I cannot trust their comparison is fair. The only detail provided is in Footnote 5, indicating that they sample 50% expert and 50% agent in their mini-batches.

Overall, I think the current work does not offer enough evidence and details to support its claims, and I cannot recommend its publication in this current form

---

> ### Author Response · Authors · 2018-11-13
> **Response to Reviewer 1**
>
> Thank you for the thoughtful review. We have addressed each of your suggestions, and added the following experiments: comparisons between SQIL and GAIL on the MuJoCo Humanoid and HalfCheetah tasks in Section 5.4, and on image-based Atari Pong in Section 5.3. We believe that this addresses all concerns, but we are happy to respond to any other suggestions you might have. We look forward to your response.
>
> > 3.	How would you make sure that the agent even starts encountering rewarding states? Do you need deterministic environments where this is more likely to happen? Do you need conditions on how much of the space is spanned by the expert demonstrations?
>
> We may not have communicated our algorithm clearly, and we’ve now revised the paper. SQIL uses off-policy RL (e.g., Q-learning) to train the agent, which allows us to directly load the expert demonstrations into the agent’s experience replay buffer. That way, the agent gets to experience rewarding states and actions, even if it never encounters them during its own interactions with the environment, because they are present in the off-policy replay buffer. The same would not be true for on-policy RL algorithms like TRPO, so the use of off-policy RL is key to our method. We have highlighted this in the abstract and Sections 1 and 3.
>
> > 1.	The authors support their method by saying that extending GAIL and other imitation learning algorithms to support pixel observations has failed...
>
> The improvements proposed in Li et al. (2017) are orthogonal to those proposed in our work, and could be combined. However, we believe that the performance of pure GAIL in our experiments is an accurate reflection of the difficulty that this method has with image observations, when not provided with auxiliary signals such as reward functions or pre-trained visual features -- Li et al. (2017) use pre-trained ImageNet features, and augment the GAIL loss with hard-coded reward functions that penalize collisions and driving off the road. We have expanded the related work paragraph in Section 1 to include Li et al. (2017).
>
> > 2.	More problematic, I think there is a clear flaw with this algorithm...
>
> The reviewer raises a good point. Since the number of demonstrations is fixed and the number of additional experiences grows over time, we balanced the number of demonstration experiences and additional experiences sampled for each gradient step in line 4 of Algorithm 1. This was not explained clearly in the original paper, and has now been clarified in Section 3.3. As the imitation policy learns to behave more like the expert, the state-action distributions of D_samp and D_demo grow closer, which causes the reward for taking the expert action in an expert state to decay from r to an expected value of r/(1 + lambda_samp). This reward decay has the effect of making the imitation policy more stochastic over time, and could potentially be countered by decreasing lambda_samp to zero over time, or decreasing the temperature parameter of the stochastic policy, but we do not find either of these fixes to be necessary in our experiments. We have added this discussion to Section 3.3.
>
> > 5.	I am not convinced of the usefulness of the lengthy derivation, although I could not follow it deeply. Especially given that the lower bound they arrive at does not seem to accurately reflect the mismatch in distributions as explained above.
>
> We have revised Sections 2 and 3 to improve clarity. The analysis provides a theoretical justification for SQIL, as well as a more detailed characterization of what SQIL is actually doing, beyond the procedural details in Algorithm 1. Section 3.1 shows that approximate MaxCausalEnt inference can be performed by optimizing the regularized behavioral cloning loss in Equation 15, and Section 3.2 shows that this regularized behavioral cloning procedure is in fact equivalent to SQIL. Thus, SQIL can be interpreted as performing approximate MaxCausalEnt inference.
>
> The reviewer raises a good point about the potential mismatch between the agent’s and the expert’s state-action distributions. Our analysis is limited in that does not show that SQIL exactly recovers the expert policy in the limit of having infinite demonstrations, or that the optimal imitation policy's state-action occupancy measure converges to the expert's. We have highlighted this limitation in Section 5.
>
> > 6.	There are no details about the network architecture used...
>
> We have added implementation details to Section A.2 in the appendix, and anonymously released code (see Section A.1 in the appendix) to encourage transparency and reproducibility. To ensure fair comparisons, we used the same network architectures and demonstration data to train SQIL, GAIL, and BC.
>
> > 4.	Additionally, Figure 1 indicates BC and GAIL as regions, without learning curves, is that after training?
>
> Yes. We have clarified this in the caption for Figure 2, and updated Figure 1 to show the learning curve for GAIL.

---

> > ### Comment · AnonReviewer1 · 2018-11-26
> > **Response**
> >
> > Thank you for your work in addressing my comments.
> >
> > I appreciate the extra work and experiments put into the paper, however I think my point 2 is still problematic enough to prevent me from increasing my score above 5.
> > The proposed solution of reducing lambda_samp does help alleviate my concern, but is not solving the real issue (the variance of the reward will still be large and affect gradients). It will revert the policy back to BC-like behaviour, which is shown to break performance in Table 3.
> >
> > I think more work and experiments are needed to really demonstrate what happens late in training, and show that the performance does not deteriorate significantly, or that the authors can find a schedule of lambda_samp which solves this issue (a better approach would be to actually find a theoretical basis for this parameter, understandably hard however).

---

> > > ### Author Response · Authors · 2018-11-27
> > > **Response to Point 2**
> > >
> > > Thank you for taking the time to read our response and update your review.
> > >
> > > The reviewer raises a good point: during the late stages of training, the variance of the implicit reward for taking the expert action in an expert state may increase. We would like to alleviate this concern (1) empirically and (2) theoretically.
> > >
> > > (1) We have added a new experiment (see Figure 4 in the appendix) in which we increase the number of training steps for SQIL from 200k steps to 500k steps on the Humanoid task from MuJoCo, which shows that training for a much longer period of time does not lead to significant deterioration in performance. Together with the training curves from the Car Racing experiment (see Section 5.2) and the Pong experiment (see Section 5.3), these results suggest that SQIL's performance does not necessarily deteriorate during the late stages of training. We will add additional experiments that train the agent for larger numbers of training steps in the final paper. We would like to emphasize that we did not change the value of lambda_samp or the temperature of the stochastic policy over time, and kept them constant during training.
> > >
> > > (2) Even though the implicit reward for taking the expert action in an expert state may decay from r to r/(1 + lambda_samp), this reward remains positive, which means the agent is still encouraged to imitate the expert demonstrations. Balancing the number of demonstration experiences and additional experiences sampled for each gradient step ensures that this implicit reward remains positive. Changing lambda_samp or the temperature is not necessary.
> > >
> > > Is there anything else we can do to address point 2, that would affect your overall rating of the paper?

---

> ### Author Response · Authors · 2018-11-22
> **Request for Feedback**
>
> Thank you again for the thoughtful review. We would like to know if our rebuttal (see below, "Response to Reviewer 1") adequately addressed your concerns. We would also appreciate any additional feedback on the revised paper. Are there any other aspects of the paper that you think could be improved?

---

### Official Review · AnonReviewer3 · 2018-11-05
**An interesting approach for imitation learning, but need to verify the method thoroughly**

**Rating:** 5
**Confidence:** 4

**Review:**

In this paper, the author derived a unified approach to utilize demonstration data and the data collected from the interaction between the current policy and the environment, inspired from soft Bellman equation. Different from previous methods such as DQfD (Hester et al., 2017), SQIL does not require the reward signal of the expert data, which is more general and natural for real-world applications such as demonstration from the human.  The author verified SQIL on a toy Lunar Lander environment and a high-dimension image based observation environment, which demonstrate its advantages over behavior cloning and GAIL. Besides the advantages, I have serval concern which may help the author to further improve the paper.

- The core algorithm is simple but I found the derivation is hard to read, which is a little messy from equation (5) to (7), and also the final loss (14) seems to be unrelated to the previous derivations( from equation (11) to (13)).  Also, can you add the gradient updates for $\theta$ with equation (11) (denoted by SQIL-11 in your paper)? I am looking forward to reading the revised version.

- To demonstrate the advantages of SQIL in high-dimension observations, the author only conducts one simple environment Car Racing, which is not enough to demonstrate its advantages. I wonder if it is possible to run more benchmark environments such as Atari game or Minecraft.

- In the related work, the author argues that methods such as DQfD require reward signal, but it would be great to demonstrate the advantages of SQIL over these methods (including DQfD and NAC (Gao et.al, 2018)).

- In previous imitation methods such as GAIL, they studied the effect of the amount of the demonstration data, which the paper should also conduct similar experiments to verify the advantage of SQIL.


Hester, Todd, et al. "Deep Q-learning from Demonstrations." arXiv preprint arXiv:1704.03732 (2017).
Gao, Yang, et al. "Reinforcement learning from imperfect demonstrations." arXiv preprint arXiv:1802.05313 (2018).

---

> ### Author Response · Authors · 2018-11-13
> **Response to Reviewer 3**
>
> Thank you for the thoughtful review. We have addressed each of your suggestions, and added the following experiments: comparisons between SQIL and GAIL on the MuJoCo Humanoid and HalfCheetah tasks in Section 5.4, and on image-based Atari Pong in Section 5.3. SQIL significantly outperforms GAIL on image-based Pong, and matches the performance of GAIL on the low-dimensional MuJoCo tasks. We believe that this addresses all concerns, but we are happy to respond to any other suggestions you might have. We look forward to your response.
>
> > To demonstrate the advantages of SQIL in high-dimension observations, the author only conducts one simple environment Car Racing, which is not enough to demonstrate its advantages. I wonder if it is possible to run more benchmark environments such as Atari game or Minecraft.
>
> We have added a preliminary experiment on image-based Atari Pong to Section 5.3. The results suggest that SQIL outperforms GAIL -- in particular, a version of GAIL that uses Q-learning instead of TRPO. This experiment provides a head-to-head comparison of SQIL and GAIL: both algorithms use the same underlying RL algorithm (Q-learning) but provide the agent with different rewards -- SQIL provides constant rewards, while GAIL provides learned rewards. The results show that, as in the image-based Car Racing game, GAIL struggles on image-based Pong to learn a policy that is better than random. SQIL does not match the expert either, but achieves substantial improvement over the random policy. We will run additional experiments on other Atari games for the final version of the paper.
>
> > In previous imitation methods such as GAIL, they studied the effect of the amount of the demonstration data, which the paper should also conduct similar experiments to verify the advantage of SQIL.
>
> We have added MuJoCo experiments to Section 5.4, in which we vary the amount of demonstration data. Both SQIL and GAIL generally achieve good performance with a very small number of demonstrations. These tasks are relatively simple, and GAIL already performs optimally on these tasks, so there is little room for improvement. Hence, the focus of our experiments is to test the generalization capabilities of SQIL and GAIL, rather than their sample efficiency with respect to demonstrations.
>
> > The core algorithm is simple but I found the derivation is hard to read, which is a little messy from equation (5) to (7), and also the final loss (14) seems to be unrelated to the previous derivations( from equation (11) to (13)).  Also, can you add the gradient updates for $\theta$ with equation (11) (denoted by SQIL-11 in your paper)? I am looking forward to reading the revised version.
>
> We have revised the analysis in Sections 2 and 3 to improve clarity. Section A.1 of the appendix now contains a more detailed description of SQIL-T (formerly "SQIL-11").
>
> > In the related work, the author argues that methods such as DQfD require reward signal, but it would be great to demonstrate the advantages of SQIL over these methods (including DQfD and NAC (Gao et.al, 2018)).
>
> The reviewer raises a good point. DQfD and NAC are demonstration-accelerated RL algorithms, which require knowledge of the reward function, while SQIL is an imitation learning algorithm that does not. Hence, when a reward signal is observed, SQIL has no advantage over DQfD or NAC, just as pure imitation learning has no advantage over demonstration-accelerated RL. We have added a footnote to page 5 to clarify this.

---

> ### Author Response · Authors · 2018-11-22
> **Request for Feedback**
>
> Thank you again for the thoughtful review. We would like to know if our rebuttal (see below, "Response to Reviewer 3") adequately addressed your concerns. We would also appreciate any additional feedback on the revised paper. Are there any other aspects of the paper that you think could be improved?

---

### Public Comment · (anonymous) · 2018-10-16
**Some Questions on Math**

Thank you for your interesting paper. I enjoyed reading it. However, I had some difficulty in digesting some math.

First, how is (6) obtained? The first inequality is a result of Jensen's inequality. At first sight, I thought Jensen's inequality was applied again. However, if so, isn't the lower bound E_{R,Q} [ \log p(t|Q)]?

Second, (6) contains \log p(Q|R). Later, it is defined as a delta function. A delta function is not a function. Rather, it is a distribution (or generalized function). So, I thought it is not allowed to take log, is it?

Third, (9) is \Pi_{s, a}. Here, the state space is continuous. How is \Pi_{s, a} f (s, a) defined in this case?

Fourth, (9) defined P (Q|R) as \lim_{\sigma \rightarrow \infty} \Pi_{s, a} N (\delta_{Q, R} (s, a); 0; \sigma^2). I cannot digest this. Let us consider a simple case where there are only one state and action. Then, \delta_{Q, R} (s, a) = Q - R - \gamma Q = (1 - \gamma) Q - R. Thus, N (\delta_{Q, R} (s, a); 0; \sigma^2) = N ( (1 - \gamma) Q - R; 0; \sigma^2). Because \int P (Q|R) dQ = \int N ( (1 - \gamma) Q - R; 0; \sigma^2) dQ is not 1, it is unreasonable. If there are more than one state and action, I am not sure what will happen.

Fifth (this is not about math), there is Var[R(s, a)] in (11). In my understanding, Var[R(s, a)] = \int R (s, a)^2 p(R) dR, where p(R(s, a)) is a prior distribution of R(s, a) (E[R(s, a)] is assumed to be 0 as you did). Here, Var[R(s, a)] may be arbitrarily large as the prior distribution is arbitrary. Thus, there may be a huge gap between log p(\tau) and the lower bound (11). Isn't it an issue?

Sixth, in (12), a lower bound of (11) is established using V(s_0) \leq T (R_{max} + log |A|). It is written that the proposed algorithm can be extended to MDPs with continuous action spaces. However, when an action space is not bounded (e.g., one dimensional Euclid space), log |A| is infinity. Thus, (12) becomes a meaningless bound: (11) \geq - \infty.  Therefore, I think, the proposed algorithm can be extended to MDPs with "bounded" continuous action spaces.

Lastly (this is not about math), again in (12), a lower bound of log p(\tau) is established using V(s_0) \leq T (R_{max} + log |A|). Depending on an action space, log |A| may become huge. As a result, there may be a huge gap between log p(\tau) and the lower bound (12). Isn't it an issue?

---

> ### Author Response · Authors · 2018-10-24
> **Revisions in Response to OP's Comment**
>
> Thank you for the thoughtful feedback. We’ve rewritten Sections 2 and 3 to address your comments. The overall derivation still holds -- it shows that SQIL maximizes a lower bound on the log-posterior of the soft Q values, and that this lower bound trades off between a behavioral cloning loss and a regularization term that incorporates state transition information -- but we have worked on clarifying our assumptions, notation, and proof steps. We will post the revised draft once the rebuttal period begins on November 8.
>
> > First, how is (6) obtained? The first inequality is a result of Jensen's inequality. At first sight, I thought Jensen's inequality was applied again. However, if so, isn't the lower bound E_{R,Q} [ \log p(t|Q)]?
>
> Good point. We have reformulated the problem as optimizing the posterior probability of the soft Q values, instead of optimizing the marginal probability of the demonstrations. This approach avoids the second inequality used to obtain (6).
>
> > Second, (6) contains \log p(Q|R). Later, it is defined as a delta function. A delta function is not a function. Rather, it is a distribution (or generalized function). So, I thought it is not allowed to take log, is it?
>
> Good point. In the revised draft, we will present our approximation to the MaxCausalEnt model upfront in Section 2. In the standard MaxCausalEnt model, soft Q values are a deterministic function of the rewards and dynamics, given by the soft Bellman equation, Q(s,a) := R(s,a) + \gamma V(s’). In our approximation, the soft Q values are treated as random variables, Q(s,a) ~ Normal(R(s,a) + \gamma V(s’), \sigma^2), where \sigma is a constant hyperparameter. With this approach, we no longer invoke the delta function in the analysis.
>
> > Third, (9) is \Pi_{s, a}. Here, the state space is continuous. How is \Pi_{s, a} f (s, a) defined in this case?
>
> By reformulating the problem as optimizing the posterior probability of a finite set of soft Q values, we avoid this product over a potentially infinite set of continuous states. In particular, we only consider the finite set of soft Q values \bm{Q} := {Q(s,a) | (s,a) in X}, where X is a finite subset of the state-action space that contains the state-action pairs in the demonstrations. This will be clarified in the revised draft.
>
> > Fourth, (9) defined P (Q|R) := \lim_{\sigma \rightarrow \infty} \Pi_{s, a} N (\delta_{Q, R} (s, a); 0; \sigma^2). I could not digest this. Let us ignore \lim. The right hand side must satisfy \int \Pi_{s, a} N (\delta_{Q, R} (s, a); 0; \sigma^2) dQ = 1. However, it is not obvious if the condition holds (note that \delta_{Q, R} involves not only Q(s, a) but also Q(s', a')).
>
> In the revised draft, we will ensure that the joint distribution over soft Q values p(\bm{Q} | R) can be factored as \Pi_{(s,a) in X} Normal(Q(s,a); R(s,a) + \gamma V(s’), \sigma^2), by making two assumptions: that no state can be entered more than once per episode (ensuring that the probabilistic graphical model that captures the joint distribution over soft Q values is acyclic), and that X satisfies the closure property, forall (s, a) in X. (s' not terminal implies (forall a' in A. (s', a') in X)).
>
> > Fifth (this is not about math), there is Var[R(s, a)] in (11). In my understanding, Var[R(s, a)] = \int R (s, a)^2 p(R) dR, where p(R(s, a)) is a prior distribution of R(s, a) (E[R(s, a)] is assumed to be 0 as you did). Here, Var[R(s, a)] may be arbitrarily large as the prior distribution is arbitrary. Thus, there may be a huge gap between log p(\tau) and the lower bound (11). Isn't it an issue?
>
> We implicitly assume that R(s,a) has bounded variance. We will add this assumption to the revised draft.
>
> > Sixth, in (12), a lower bound of (11) is established using V(s_0) \leq T (R_{max} + log |A|). It is written that the proposed algorithm can be extended to MDPs with continuous action spaces. However, when an action space is not bounded (e.g., one dimensional Euclid space), log |A| is infinity. Thus, (12) becomes a meaningless bound: (11) \geq - \infty.  Therefore, I think, the proposed algorithm can be extended to MDPs with "bounded" continuous action spaces.
>
> The lower bound can still be applied when the action space is continuous, as long as the policy \pi(a|s) has an entropy that is bounded above by some constant, where \pi is any policy in the policy class being optimized over. We will clarify this in the revised draft.
>
> > Lastly (this is not about math), again in (12), a lower bound of log p(\tau) is established using V(s_0) \leq T (R_{max} + log |A|). Depending on an action space, log |A| may become huge. As a result, there may be a huge gap between log p(\tau) and the lower bound (12). Isn't it an issue?
>
> You are correct to point out that this lower bound becomes looser as the action space grows larger. So far, we have only run experiments on environments with small, discrete action spaces, so this question merits future work.

---

> > ### Public Comment · (anonymous) · 2018-10-25
> > **Thank you for your reply.**
> >
> > I am looking forward to reading the revised paper!

---

> > ### Public Comment · (anonymous) · 2018-11-19
> > **Some more comments on maths**
> >
> > Thank you for the updates. I read it and write some comments.
> >
> > 1.  In (6), only one expert rollout is used. However, in (15), more rollouts are used. Thus, I think p(Q|t) in (6) should be p(Q|t_1, t_2, ..., t_n). In this case, t_i may contain a state transition from s to s' such that s' is in other rollouts. Isn't that an issue for p(Q|R) to be a probability distribution?
> >
> > 2. (13) shows that as T becomes larger, the lower bound gets looser. I would use (f (\sigma) + \log |A|) / (1 - \gamma).
> >
> > 3. (13) is derived considering p(Q|t), where t is expert's rollout. However, (15) contains rollouts of the agent. How are rollouts of the agent related to (13)?

---

> > > ### Author Response · Authors · 2018-11-20
> > > **Response to Comments on Sections 2 and 3**
> > >
> > > Thank you again for the detailed feedback. We have uploaded a revised draft that addresses your comments.
> > >
> > > > 1.  In (6), only one expert rollout is used. However, in (15), more rollouts are used. Thus, I think p(Q|t) in (6) should be p(Q|t_1, t_2, ..., t_n). In this case, t_i may contain a state transition from s to s' such that s' is in other rollouts. Isn't that an issue for p(Q|R) to be a probability distribution?
> > >
> > > Good point. To fix this issue, we have strengthened the assumption about the dynamics in Section 2, which now constrains the dynamics in the following way: if state s_i is reachable from s_j, then s_j is not reachable from s_i.
> > >
> > > > 2. (13) shows that as T becomes larger, the lower bound gets looser. I would use (f (\sigma) + \log |A|) / (1 - \gamma).
> > >
> > > We have updated the bound accordingly.
> > >
> > > > 3. (13) is derived considering p(Q|t), where t is expert's rollout. However, (15) contains rollouts of the agent. How are rollouts of the agent related to (13)?
> > >
> > > We have added a footnote to page 4 explaining that augmenting the demonstrations with rollouts of the partially-trained imitation policy improves our approximation of the objective in Equation 14, since the demonstrations alone may not satisfy the closure property in Equation 5. That being said, even with the added rollouts, Equation 5 is not guaranteed to be satisfied. The experiments in Section 5 suggest that this is not an issue empirically.

---

### Meta-Review · Area_Chair1 · 2018-12-14
**Not quite there yet.**

**Confidence:** 4
**Recommendation:** Reject

**Metareview:**


This is an interesting direction and all reviewers thought the idea has merit, but pointed out some significant limitations. The authors did an admirable job at addressing some of these but some remain, including R1’s point 2 which is a significant issue. The authors are encouraged to submit a revised version of their work which addresses all the discussed limitations and will likely be a competitive submission to another top ML conference.